# Vgent: Graph-based Retrieval-Reasoning-Augmented Generation For Long Video Understanding

**Xiaoqian Shen**[1], **Wenxuan Zhang**[1], **Jun Chen**[1,2], **Mohamed Elhoseiny**[1]
[1]King Abdullah University of Science and Technology, [2]Meta AI
{xiaoqian.shen,wenxuan.zhang,jun.chen,mohamed.elhoseiny}@kaust.edu.sa

## Abstract

Understanding and reasoning over long videos pose significant challenges for large video language models (LVLMs) due to the difficulty in processing intensive video tokens beyond context window and retaining long-term sequential information. Retrieval-Augmented Generation (RAG) has demonstrated effectiveness in processing long context for Large Language Models (LLMs); however, applying RAG to long video faces challenges such as disrupted temporal dependencies and inclusion of irrelevant information that can hinder accurate reasoning. To address these limitations, we propose Vgent, a novel **graph-based retrieval-reasoning-augmented generation framework** to enhance LVLMs for long video understanding. Our approach introduces two key innovations: (i) It represents videos by structured graphs with semantic relationships across video clips preserved to improve retrieval effectiveness. (ii) It introduces an intermediate reasoning step to mitigate the reasoning limitation of LVLMs, which leverages structured verification to reduce retrieval noise and facilitate the explicit aggregation of relevant information across clips, resulting in more accurate and context-aware responses. We comprehensively evaluate our framework with various open-source LVLMs on three long-video understanding benchmarks. Our approach yielded an overall performance improvement of $3.0\% \sim 5.4\%$ over base models on MLVU, and outperformed state-of-the-art video RAG methods by $8.6\%$. Our code is publicly available at https://xiaoqian-shen.github.io/Vgent.

## 1 Introduction

Multi-Modal Large Language Models (MLLMs) [6, 22, 30, 39, 40, 58] have demonstrated remarkable visual understanding and reasoning capabilities, paving the way for advancements in video tasks. Recently, numerous studies have showcased impressive progress in building Large Video Language Models (LVLMs) for video understanding [9, 7, 24, 35, 55]. Moreover, long-video understanding is particularly crucial for applications in web content, life-logging, and streaming media, where intricate narratives and evolving contexts span extended durations.

However, processing and reasoning over long-context videos remain a formidable challenge for existing LVLMs, as representing video frames requires an extensive number of tokens—for example, a 30-minute video can exceed 200K tokens [4, 22], beyond most models' context limits. To handle longer videos, existing methods resort to sparse frame sampling [55, 22] or token compression [43], but these approaches inevitably lead to visual information loss, weakening fine-grained temporal understanding and coherent reasoning.

Recent studies [1, 2, 12, 34, 52, 53] utilize Retrieval-Augmented Generation (RAG) [21] to enhance long-form video understanding by retrieving relevant information. However, they encounter certain limitations. First, some works segment lengthy videos into shorter clips, treating each as an individual document for retrieval [2], which disrupts the continuity of entities and temporal dependencies,

39th Conference on Neural Information Processing Systems (NeurIPS 2025).

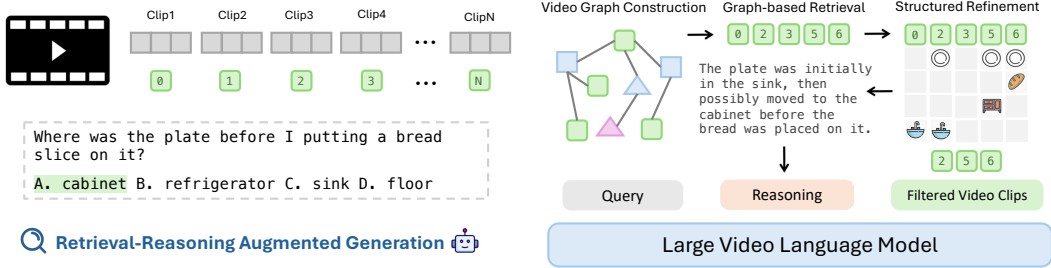

Figure 1: **Overview of our graph-based retrieval-reasoning-augmented generation framework.** Each video clip is represented as a node within a graph, interconnected through shared entities. This graph representation enables effective retrieval of relevant clips based on node connections, followed by an intermediate reasoning step to refine retrievals and aggregate over multimodal context for accurate generation.

leading to retrieval inaccuracies. Second, some methods [47, 34] rely on proprietary LLMs like GPT-4 [37] for multi-turn interactions, planning, and reasoning, making them costly and less flexible. Lastly, several approaches [47, 33] extract information from sparse key frames, neglecting temporal coherence in long videos.

**Graph representation for enhanced retrieval**: To address the limitation of existing RAG methods, we propose a structured graph-based representation, where video clips are modeled as nodes interconnected by recurring subjects or scenes. This graph representation not only enables effectively retrieving nodes associated with specific entities, but also facilitates capturing temporal dependencies spanning over lengthy videos. Another advantage is that the graph construction is performed offline and is query-independent. Once the graph is built, it can be reused for multiple questions on the same video, allowing retrieval to operate directly on the graph without reprocessing the video. However, feeding all retrieved clips into LLMs can cause information overload, where key details are diluted by irrelevant content [14]. In videos, the issue is further amplified, as each frame consumes hundreds of tokens, with irrelevant information overshadowing the critical content.

**Structured post-retrieval reasoning**: To address the aforementioned issue and fully harness the benefits from our GraphRAG, we introduce the structured reasoning step in the post-retrieval stage. Instead of generating answers directly from the retrieved clips, it decomposes the question and systematically verifies the relevance of each clip. As shown in Figure 1, this process refines the retrieved set by identifying clips that mention critical elements—such as the plate, sink, cabinet, and bread—and then aggregates information across them for temporal reasoning (e.g., the plate moving from the sink to the cabinet). This process mitigates noise, facilitates information aggregation across refined clips, and thus creates a more reliable pathway for producing accurate responses.

We evaluate our framework upon seven different LVLMs with sizes ranging from 2B to 7B across three long-video benchmarks: MLVU [57], VideoMME [13] and LongVideoBench [50]. Experimental results demonstrate that our framework consistently improves the performance of existing LVLMs by 3.0%–5.4%. We further show that our framework surpasses existing RAG-based video understanding works by 8.6%.

**Contribution.** Our contribution is summarized as follows:

- We developed a novel *graph-based RAG* framework for long-video understanding, where video clips are represented as nodes within a graph, interconnected through shared entities, thereby preserving semantic relationships and temporal dependencies across clips, facilitating more effective retrieval.

- We propose *structured reasoning* to tackle the limited reasoning ability of LVLMs, which can be distracted by hard negative retrieved samples. Our approach introduces an intermediate reasoning step for retrieval verification and aggregates information across verified clips to enhance generation accuracy.

- Our graph-based retrieval-reasoning-augmented framework demonstrates 3.0%–5.4% improvements over various LVLMs ranging from 2B to 7B and surpasses existing RAG-based video understanding works by 8.6% in long-video understanding tasks.

## 2 Related Work

### 2.1 Large Video Language Models

Multimodal large language models (MLLMs) [6, 58, 31, 30, 45] have demonstrated remarkable progress in vision-language tasks. Recent advancements have further extended their capabilities to video understanding tasks [2, 9, 24, 25, 28, 32]. Large Video Language Models (LVLMs) process videos by extracting and encoding frames, and then rearranging them into final video representations. Some approaches [24, 25, 9] leverage the Q-Former module from BLIP-2 [23] to integrate visual and textual features, while others [2, 28, 32] directly concatenate frame features. However, these models struggle with processing hour-long videos in a single pass as the number of video tokens exceeds their training context size. To address these limitations, most existing works train on sparsely sampled frames no matter how long the video is [24, 2, 9, 55, 22], while others try to handle long videos by token pooling [35, 27, 44], token compression [43], or memory aggregation [19]. However, they struggle to effectively capture and reason about temporal dependencies spanning hour-long videos.

### 2.2 Agent-based Video Understanding

A dominant trend in long-context video question-answering involves equipping Large Language Models (LLMs) with tools that heavily rely on proprietary models to process queries and handle video clips. MM-VID [29] uses a video-to-script generation with GPT-4V [37] to transcribe multimodal elements into a long textual script. VideoAgent [47] integrates diverse foundation models through a unified memory architecture. DrVideo [34], VideoTree [49], VideoAgent [12] and OmAgent [53] dynamically invoke tools to enhance query processing and accuracy. Such methods consequently suffer from high operational costs and a critical reliance on external, closed-source systems, limiting their adaptability. In contrast, our work targets the development of a self-contained pipeline designed for flexible deployment with open-source LVLMs.

### 2.3 Video Retrieval-Augmented Generation

Retrieval-Augmented Generation (RAG) enhances large language models (LLMs) by retrieving relevant information to improve long context memory, factual accuracy, and reduce hallucinations. The process involves three stages: (i) indexing, which organizes raw data into a knowledge base; (ii) retrieval, which searches for relevant information based on user queries; and (iii) generation, where the model takes the retrieved context to generate the final response. Recent advancements of RAG in LLM mainly follow two directions, i.e. chunk-based methods [15, 5] and graph based methods [11, 17, 26, 18], both of which have been applied in video understanding tasks. Goldfish [3] chunks long videos into shorter clips, processes each clip independently, and retrieves the most relevant clip in response to user queries. Wang et al. [48] applied graph structures for action recognition in short clips and Hussein et al. [20], Luo et al. [33] employ scene graphs for video understanding. However, constructing graphs for long videos and effectively retrieving information from the noisy and complex graph remains a challenge. Only recently, a concurrent work [42] constructs graphs for long-context video understanding, but they heavily relies on external proprietary LLM for graph construction, while graph-based video RAG with open-sourced LVLM itself remains unexplored, which our work aims to address.

## 3 Method

We introduce a novel, training-free framework, Vgent, for long-context video understanding. Unlike conventional Retrieval-Augmented Generation (RAG), our pipeline proposes a **graph-based retrieval-reasoning-augmented generation** paradigm, specifically designed to address complex video scenarios with improved contextual comprehension and structured reasoning. As illustrated in Figure 2, our proposed pipeline contains four stages: (1) Offline video graph construction (Section 3.1): Builds a video graph offline by extracting knowledge from long videos. (2) Graph-based

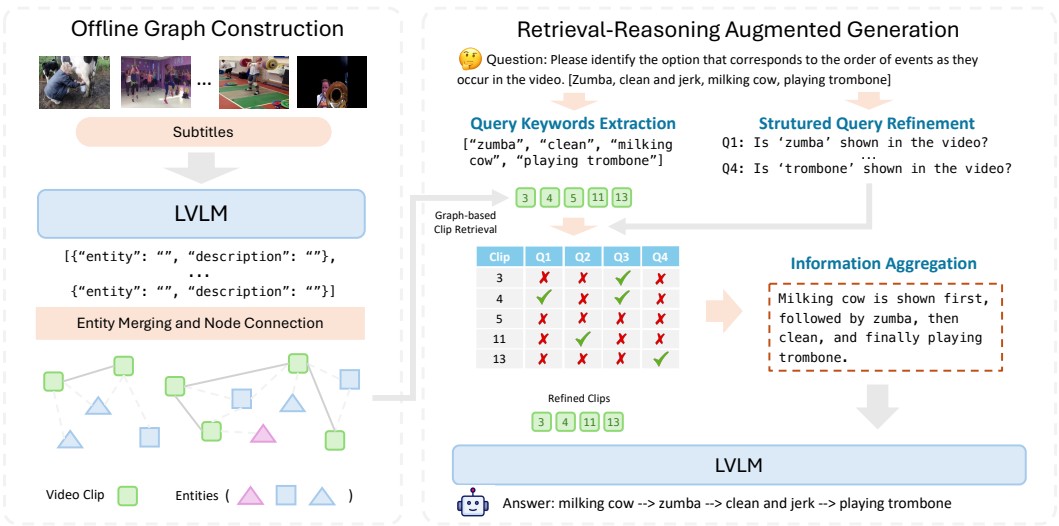

Figure 2: Pipeline of **Vgent**, a novel framework for long-context video understanding in the proposed graph-based retrieval-reasoning-augmented generation paradigm. It consists of four key stages: (1) Offline video graph construction (Section 3.1): Builds a video graph by extracting knowledge from long videos. (2) Graph-based retrieval (Section 3.2): Retrieves relevant clips based on keywords extracted from the user query. (3) Structured reasoning (Section 3.3): Refines clips using structured queries and aggregates information. (4) Multimodal augmented generation (Section 3.4): Combines refined clips and reasoning results to generate the final response.

retrieval (Section 3.2): Retrieves relevant video clips from graph based on the user query. (3) Structured Reasoning (Section 3.3): Refines the retrieved clips using structured queries and aggregates information across the filtered clips. (4) Multimodal Augmented Generation (Section 3.4): Combines refined clips and intermediate reasoning results to generate the final response.

## 3.1 Video Graph Construction

To better capture the complex relationships and dependencies in long-context videos, we propose a graph-based representation to store video content and enhance semantic connections. Specifically, given a video $V$ with $F$ frames, we first partition it into a sequence of short video clips $\{V_1, V_2, \ldots, V_{\lceil \frac{F}{K} \rceil}\}$, where each video clip $V_i$ consists of $K$ frames. We then dynamically construct the graph by a series of structured steps, as detailed below.

**Visual Entity Extraction.** For each video clip, we leverage the LVLM to extract the key semantic entities (i.e., the primary subjects, actions, or scenes) from both the spoken content (subtitles) $C_i$ and video clip $V_i$.

$$\{(e_1^i, t_1^i), (e_2^i, t_2^i), ...\} \leftarrow \texttt{LVLM}\,(C_i, V_i), \tag{1}$$

where the set of entity is denoted as $E_i = \{e_i^1, e_i^2, \ldots\}$ and its corresponding description set is denoted as $T_i = \{t_i^1, t_i^2, \ldots\}$. In this step, the LVLM captures subjects, actions, and scene dynamics, seamlessly linking visual entities with spoken content to extract meaningful knowledge. Please refer to Appendix B.1 for illustrative examples.

**Graph Construction.** Based on extracted information, we construct a video knowledge graph $\mathcal{G} = (\mathcal{V}, \mathcal{E})$, where $\mathcal{V}$ denotes the nodes set representing video clips, and edges in $\mathcal{E}$ represents the connectivity between nodes. Additionally, we define a global set of unique prototype entities $\mathcal{U} = \{u \in E_i, i = 1, \ldots, \lceil \frac{F}{K} \rceil\}$ that spans across all nodes. As more video clips are processed, we will dynamically add newly extracted unique entity $u$ to the set or link it to an existing entity. We define $t^u$ as the description of each entity $u \in \mathcal{U}$.

**Entity Merging and Node Connection.** Since LVLMs process video clips independently, it is essential to identify and unify semantically equivalent entities across clips. Given a newly extracted entity-description pair $(e_i^j, t_i^j)$ from video clip $V_i$, we determine whether it belongs to an existing entity in the global entity set $\mathcal{U}$. Specifically, we compute the similarity score between the textual descriptions $t_j^i$ and descriptions of entities in $\mathcal{U}$ based on their respective text embeddings. If the similarity score $> \tau$, the entity $e_j^i$ is considered semantically equivalent to an existing entity and these two are merged into a single entity representation. Otherwise, $e_j^i$ is treated as a distinct entity and added to $\mathcal{U}$. This process is formulated as follows:

$$s^* = \max_{u \in \mathcal{U}} sim(t_j^i, t), \ \ u^* = \arg\max_{u \in \mathcal{U}} sim(t_j^i, t^u), \ \ e_j^i \rightarrow \begin{cases} u^*, & \text{if } s^* \geq \tau \\ \mathcal{U} \leftarrow \mathcal{U} \cup \{e_j^i\}, & \text{otherwise} \end{cases} \quad (2)$$

Once entity is merged, we then build edges from the node $v_i$ associated with the video clip $V_i$ to all the nodes that have the same entity $u^*$, denoted as $\mathcal{V}^{(u^*)}$.

$$\mathcal{E} \leftarrow \mathcal{E} \cup \{(v_i, v) \mid v \in \mathcal{V}^{(u^*)}\} \quad (3)$$

As new video clips are processed, the graph is dynamically updated such that nodes containing the same entity are connected, which preserves semantic relationships and contextual dependencies. This forms a structured representation that facilitates effective video retrieval in subsequent processing stages.

## 3.2 Graph-based Retrieval

**Keywords Extraction.** Direct retrieval based on the original query may not provide sufficient context, especially when reasoning across multiple temporal clips is required. To address this, we extract keywords from the query for effective retrieval. Specifically, we prompt the LVLM to identify key semantic elements, denoted as $\mathcal{K}$, from the query $Q$. The detailed prompt is provided in Appendix B.2.

**Graph-based Clip Retrieval.** Next, we leverage these extracted keywords for graph-based retrieval. Specifically, for each keyword $k \in \mathcal{K}$ and each entity $u \in \mathcal{U}$, we compute a similarity score $sim(k, t^u)$ to determine whether the entity matches the keyword. If $sim(k, t^u) > \theta$, we include all nodes associated with entity $u$ as the target retrieval node set $\mathcal{R}$:

$$\mathcal{R} = \bigcup_{u \in \mathcal{U}, k \in \mathcal{K}} \{v \in \mathcal{V} \mid u \in \mathcal{U}(v), sim(k, t^u) > \theta\} \quad (4)$$

After obtaining the retrieval node set $\mathcal{R}$, we refine the results by re-ranking the nodes based on the similarity between the query's keywords and the extracted information of each node, including entities, corresponding textual descriptions, and subtitles if available. Finally, we select the Top-$N$ nodes with the highest average similarity scores across all associated information of each video clip.

## 3.3 Structured Reasoning

Feeding all relevant clips directly into LLMs can lead to information overload, diluting the focus on key details with irrelevant content [14]. Our empirical analysis also reveals that in roughly 40% of failure cases, the correct clip is successfully retrieved, yet the model still generates incorrect responses—even though it can answer correctly when provided with that clip alone. We then introduce structured reasoning in the post-retrieval stage that refines the retrieved clips and aggregates useful information towards final generation.

**Structured Query Refinement.** We introduce the divide-and-conquer strategy to refine the retrieval through structured query verification. Specifically, we prompt the LVLM to generate structured subqueries, denoted as $\mathcal{Q}$, based on the original query $Q$ and extracted keywords $\mathcal{K}$. These subqueries focus on verifying the presence of relevant entities or quantifying their occurrences, whose answers are expected to be binary (yes/no) or numerical value. Please refer to Appendix B.3 for the detailed prompt and Figure 3 for an example of generated subqueries.

After generating the subqueries, we process the Top-$N$ retrieved video clips using the LVLM, producing either binary (yes/no) or numerical responses for each subquery. As shown in Figure 2, this structured verification systematically assesses the relevance of each clip to the original query, filtering out irrelevant clips that were wrongly retrieved based on semantic embedding similarity. Denoting 1 to `yes` and 0 to `no` in binary questions, this refined clip set $\mathcal{R}'$ can be formulated as:

$$\mathcal{R}' = \{v_i \in \mathcal{R} \mid \exists q_j \in \mathcal{Q}, \, f(v_i, q_j) > 0\} \tag{5}$$

where $f(v_i, q_j)$ denotes the response of retrieved clip $v_i$ to subquery $q_j$. We keep at most $r$ clips after refinement. This refinement step ensures that only video clips satisfying the structured queries are retained, effectively eliminating hard negatives from the initial retrieval.

**Information Aggregation.** As shown in Figure 2, we then let LVLM aggregate and summarize all useful information from structured queries and their corresponding results for each video clip, providing an enriched auxiliary context that enhances the final inference.

### 3.4 Multimodal Augmented Generation.

We incorporate both the intermediate reasoning results and the filtered video clips as multimodal context inputs to the LVLM for the final response. This enriched input allows the model to leverage both structured reasoning and relevant visual information, enabling it to generate a more accurate and contextually grounded final response to the original question.

## 4 Experiments

### 4.1 Experimental Setups

**Baselines.** We apply our framework Vgent on open-sourced LVLMs including InternVL2.5 [8], Qwen2 [46], Qwen2.5-VL [4], LongVU [43] and LLaVA-Video [56] as base video understanding model. We further compare Vgent against state-of-the-art RAG baselines as follows: **NaïveRAG** [3], **Video-RAG** [33], and **proprietary LLM-based methods** including VideoAgent [47], LLoVi [52], DrVideo [34] and VideoTree [49]. More details can be found in Appendix A.3.

**Benchmarks.** We evaluate the performances of each model across three long-video benchmarks. Video-MME [13] is a widely used benchmark designed to evaluate LVLMs' capability to process detailed, real-world videos. It comprises three subsets categorized by video length, ranging from 11 seconds to 1 hour. MLVU [57] is a long-video understanding benchmark with videos ranging from 3 minutes to 2 hours, with an average length of about 12 minutes. LongVideoBench (LVB) [50] focuses on referred reasoning tasks that require models to analyze long frame sequences. These questions depend on extensive temporal context and cannot be effectively addressed using a single frame or a small set of sparsely sampled frames.

**Implementation Details.** During the offline video graph construction, we sample videos at 1.0 FPS, segmenting the long video into clips, each containing $K = 64$ frames. We use the `BAAI/bge-large-en-v1.5` [51] embedding for similarity calculation. The entity merging threshold is set to $\tau = 0.7$. In the online retrieval stage, we use `BAAI/bge-large-en-v1.5` to retrieve the top $N = 20$ clips based on extracted keywords (maximum to 20 to discard low-relevance, with a similarity threshold $\theta = 0.5$). After structured query refinement, we retain a maximum of $r = 5$ clips. Thresholds are set as the same for all three benchmarks, with hyper-parameter selection details provided in the supplementary. For MLVU [57], we extract spoken content using `openai/whisper-large`, while for VideoMME [13] and LongVideoBench [50], we use the provided subtitles from benchmark. All experiments are conducted on A100 80G GPUs.

### 4.2 Main Results

**Comparison with LVLMs.** In Table 1 and 6, we present the performance of our proposed Vgent framework on the MLVU [57] benchmark, where we consistently observe substantial improvements across all models. Specifically, our framework enhances LongVU [43] by 5.4% and boosts Qwen2.5VL [4] (7B) by 3.3%. Notably, when applied to Qwen2.5VL (3B), Vgent achieves

Table 1: **Performance comparison with LVLMs.** Vgent consistently improves all models on MLVU [57], enhancing LongVU by 5.4% and Qwen2.5VL (7B) by 3.3%. Notably, Vgent achieves 70.4% accuracy on Qwen2.5VL (3B), surpassing its 7B counterpart and improving the base model by 4.2%. Vgent outperforms base models across all video lengths on VideoMME [13] achieving improvement of 3.2% overall.

| Models | Size | MLVU | VideoMME | | LVB |
| --- | --- | --- | --- | --- | --- |
| | | | *w/o sub.* | *w/ sub.* | |
| *Proprietary LVLMs* | | | | | |
| Gemini 1.5 Pro [16] | - | - | 75.0 | 81.3 | 64.0 |
| GPT-4o [38] | - | 64.6 | 71.9 | 77.2 | 66.7 |
| *Open-Source LVLMs* | | | | | |
| InternVL2.5 [8] | 2B | 56.7 | 49.5 | 55.2 | 52.0 |
| InternVL2.5 + Vgent (Ours) | 2B | $61.1^{+4.4}$ | $50.9^{+1.4}$ | $56.8^{+1.6}$ | $54.8^{+2.8}$ |
| Qwen2.5-VL [4] | 3B | 66.2 | 61.4 | 67.6 | 54.2 |
| Qwen2.5-VL + Vgent (Ours) | 3B | $70.4^{+4.2}$ | $63.0^{+1.6}$ | $69.6^{+2.0}$ | $57.8^{+3.6}$ |
| LongVU [54] | 7B | 65.4 | 55.2 | 60.9 | 50.2 |
| LongVU + Vgent (Ours) | 7B | $70.8^{+5.4}$ | $57.3^{+2.1}$ | $63.7^{+2.8}$ | $52.7^{+2.5}$ |
| Qwen2-VL [46] | 7B | 65.7 | 62.7 | 68.1 | 55.6 |
| Qwen2-VL + Vgent (Ours) | 7B | $70.3^{+4.6}$ | $63.5^{+0.8}$ | $70.1^{+2.0}$ | $58.4^{+2.8}$ |
| LLaVA-Video [56] | 7B | 69.5 | 64.3 | 69.2 | 59.5 |
| LLaVA-Video + Vgent (Ours) | 7B | $72.5^{+3.0}$ | $66.7^{+2.4}$ | $71.1^{+1.9}$ | $62.4^{+2.9}$ |
| Qwen2.5-VL [4] | 7B | 68.8 | 65.1 | 71.1 | 56.0 |
| Qwen2.5-VL + Vgent (Ours) | 7B | $72.1^{+3.3}$ | $68.9^{+3.8}$ | $74.3^{+3.2}$ | $59.7^{+3.7}$ |

an accuracy of 70.4%, surpassing its larger 7B counterpart and improving the base model by 4.2%. This result underscores the effectiveness of our approach in bridging the performance gap between small models and their larger counterparts. At the category level (Table 6), our framework notably improves Count and Order tasks, which demand event-level understanding and multi-clips reasoning.

In Table 1, we showcase the results of Vgent on the VideoMME [13] benchmark, where it consistently outperforms base models across all video lengths, achieving an average performance gain of 4.2%. Notably, our framework excels in long-video scenarios, surpassing the best baseline by 5.4%. These findings highlight the strength of our structured graph-based retrieval and reasoning approach, demonstrating its ability to enhance long-video comprehension by effectively capturing cross-segment dependencies and refining information retrieval for improved reasoning and final response generation.

**Comparison with SoTA RAG Methods.** In Table 2, we provide a comprehensive comparison of Vgent against state-of-the-art RAG methods on MLVU [57] and VideoMME [13] benchmarks.

(1) Our framework consistently outperforms the RAG baseline, Video-RAG [33], across three different LVLM base models. Unlike Video-RAG [33], which relies on CLIP [41]-based keyframe selection and external tools such as object detection and OCR for frame-level information extraction, Vgent eliminates these dependencies by leveraging LVLMs themselves for graph construction, verification, and intermediate reasoning. This structured approach significantly enhances retrieval precision and reasoning accuracy, leading to more reliable final responses.

(2) Our framework also surpasses proprietary RAG-based methods for long-video understanding. Compared to closed-source API-dependent methods which heavily rely on closed-source APIs, our framework is more flexible and effective solution for long-video understanding.

Table 2: **RAG methods comparison.** † denotes results are sourced from [34]. Proprietary LVLMs refer to methods that rely on closed-source APIs. We include them here for reference only, as our primary focus is on building a self-contained pipeline to improve open-source LVLMs.

| Models | Size | MLVU | VideoMME | |
| --- | --- | --- | --- | --- |
| | | | *w/o sub.* | *w/ sub.* |
| *Proprietary LVLMs* | | | | |
| VideoAgent† [47] | - | - | - | 44.4 |
| LLoVi† [52] | - | - | - | 67.7 |
| DrVideo† [34] | - | - | - | 71.7 |
| *Open-Source LVLMs* | | | | |
| Qwen2.5-VL + Video-RAG [33] | 3B | 62.2 | 60.3 | 65.1 |
| Qwen2.5-VL + Vgent (Ours) | 3B | 70.4 | 63.0 | 69.6 |
| LLaVA-Video + Video-RAG [33] | 7B | 71.3 | 64.8 | 70.0 |
| LLaVA-Video + Vgent (Ours) | 7B | **72.5** | 66.7 | 71.1 |
| Qwen2.5-VL + Video-RAG [33] | 7B | 63.4 | 60.5 | 65.7 |
| Qwen2.5-VL + Vgent (Ours) | 7B | 72.1 | **68.9** | **74.3** |

Table 3: **Ablation study results** of the performance improvement contributed by each component of our proposed pipeline. SR denotes our proposed structured reasoning.

| Models | MLVU | VideoMME | LongVideoBench |
| --- | --- | --- | --- |
| Qwen2.5-VL [4] | 68.8 | 71.1 | 56.0 |
| Qwen2.5-VL + NaïveRAG | 65.4 | 68.3 | 56.2 |
| Qwen2.5-VL + GraphRAG | 69.5 | 72.7 | 57.1 |
| Qwen2.5-VL + NaïveRAG + SR | 68.6 | 69.8 | 57.3 |
| Qwen2.5-VL + GraphRAG + SR (default) | **72.1** | **74.3** | **59.7** |

## 4.3 Ablation Studies

**NaïveRAG vs GraphRAG.** As shown in Table 3, integrating GraphRAG yields an average improvement of 2.9% over NaïveRAG, with a particularly notable 4.1% gain on MLVU [57]. This is because NaïveRAG's difficulty in handling complex queries that requires temporal reasoning across multiple clips, as it treats each video clip as an independent document. In contrast, our GraphRAG effectively preserves semantic relationships between clips, enabling more accurate retrieval and reasoning. By structuring video content into a graph representation, our approach addresses retrieval inconsistencies inherent in traditional RAG methods.

However, the improvement remains marginal compared to the base models. Upon checking failure cases in MLVU, we observe that in 44% of the failures, the correct clip is actually present within the model's retrieved set, which indicates that while the retrieval was successful, irrelevant retrievals still distract the model, hindering accurate responses. Consequently, a post-retrieval stage is necessary to amplify the potential of our GraphRAG by refining the retrieved nodes and improving reasoning towards more precise answers.

**Structured Reasoning (SR).** By refining retrieved nodes through intermediate reasoning with structured queries, we achieve an additional 2.6% improvement on MLVU [57] and 1.6% on VideoMME [13], resulting in an overall 3.4% average gain over the base model. This interme-

diate reasoning step decomposes complex queries into targeted sub-questions and generates binary or numerical answers. These structured response are then used to systematically filter out irrelevant clips and aggregate relevant information across clips, guiding the model toward the correct final answer. Our findings also indicate that the final improvement is contingent upon Graph-based RAG. Specifically, if SR is applied to NaïveRAG, the inherent inaccuracy of NaïveRAG's retrievals restricts the potential for significant improvement through refinement alone.

**Number of retrieval** $r$    We conduct an ablation study to examine the impact of the number of retrieved clips after structured query refinement. Table 4 presents both the overall performance and results across several MLVU [57] subcategories. Among these, Count and Order are two tasks that heavily require reasoning across multiple video clips. Count involves identifying the number of events or actions throughout an entire video, while Order requires the model to arrange multiple events in chronological sequence. $r$ represents the maximum number of video clips retained after refinement. Our findings indicate that increasing the number of retrieved clips consistently improves performance, particularly for tasks demanding multi-clip reasoning, with the highest performance observed at $r = 5$.

Table 4: The number of retrieved clips impacts performance on MLVU [57].

| #Retrieval | Count | Ego | Needle | Order | PlotQA | Anomaly | Topic | Overall |
|---|---|---|---|---|---|---|---|---|
| r=1 | 25.7 | 54.2 | 75.7 | 51.7 | 67.4 | 71.0 | 84.3 | 63.2 |
| r=2 | 40.2 | 55.6 | 78.0 | 57.1 | 69.1 | 73.5 | 87.0 | 66.9 |
| r=3 | 47.5 | 57.1 | 78.0 | 61.0 | 70.0 | 71.5 | 87.2 | 68.4 |
| r=4 | 58.7 | 56.6 | 78.8 | 65.2 | 73.6 | 72.5 | 87.6 | 71.0 |
| r=5 (default) | 58.7 | 59.5 | 79.7 | 67.1 | 74.6 | 74.0 | 88.0 | 72.1 |
| r=6 | 58.7 | 58.4 | 78.8 | 67.2 | 73.9 | 73.5 | 87.4 | 71.9 |

Further details, including category-level performance on MLVU (A.1), limitations (D), ablation studies on the number of retrievals $N$ (A.5), confidence-based refinement (A.2), retrieval threshold $\tau$ (A.6) are provided in the Appendix.

Table 5: **Inference time analysis.** Since processing time depends on the video duration, we report the normalized time required to process each minute of video.

| Model / Time (sec) | Query Independent (offline) | Query Dependent (online) |
|---|---|---|
| ***#Proprietary LVLMs*** | | |
| VideoAgent [47] | N/A | 67.25 |
| ***#Open-Source LVLMs*** | | |
| Qwen2.5VL-7B [4] | N/A | 2.95 |
| +Video-RAG [33] | N/A | 20.81 |
| + Vgent (Ours) | 20.13 | 3.93 |

## 4.4   Inference Time Analysis

We analyze the computational trade-offs and report the processing times in Table 5 for the API-based method VideoAgent [47], Video-RAG [33] as well as our framework built on Qwen2.5VL [4]. VideoAgent [47] leverages a proprietary LLM (GPT-4 [36]) to iteratively perform self-reflection for frame selection and aggregating key information from the video. Video-RAG [33] relies on query-dependent key frame selection and per-frame object detection, introducing online computational overhead. In contrast, our framework can offline constructs a query-independent graph from the video, which takes 20.13 seconds. Once the graph is built, the online retrieval, reasoning and generation process requires only 3.93 seconds per minute-video.

Our offline graph construction further improves efficiency in multi-question scenarios (e.g., three questions per video in VideoMME [13]). Unlike query-dependent methods that reprocess the entire

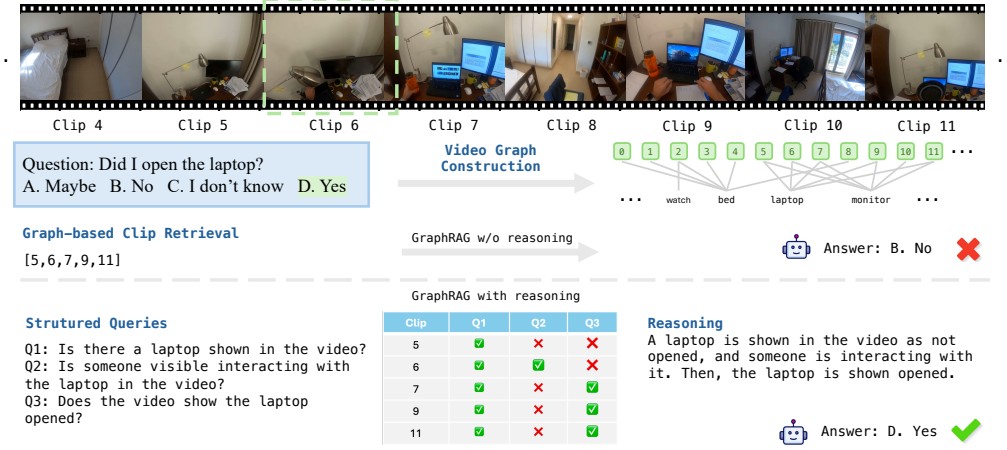

Figure 3: A qualitative example illustrates our graph-based retrieval-reasoning approach, which connects relevant video clips via shared entities. While the model initially fails to correctly identify the action of opening the laptop, misled by hallucinations from hard negatives. However, adding an intermediate reasoning step that validates each retrieved node through structured subqueries enables the model to generate the accurate response.

video for each question, our approach constructs the graph once, allowing the model to retrieve relevant clips based on entity descriptions—without the need to rewatch the entire video. As a result, our method achieves a 1.73× speedup over Video-RAG [33] when performing inference on VideoMME [13].

## 4.5 Qualitative Examples

We show a qualitative example in Figure 3, 5 and 6. Our graph construction effectively connects relevant video clips through shared entities. In Figure 3 the graph-based retrieval system can identify relevant nodes that contains a laptop, with Clip 6 providing crucial evidence to answer the query. However, the model incorrectly responded "No" to the question "Did I open the laptop?", presumably due to hard negatives from multiple clips featuring a opened laptop, hallucinating the model to overlook the closed laptop and the action of opening it.

In contrast, with an intermediate reasoning step, we validate each retrieved node with structured subqueries (e.g., "Is there a laptop open?" "Is someone interacting with the laptop?"). This verified information is aggregated to form an enhanced reasoning chain, allowing the model to correctly infer that the laptop was opened, overcoming the distraction from hard negatives.

## 5   Conclusion

In this work, we introduced a novel graph-based Retrieval-Augmented Generation (RAG) framework designed for long-video understanding. Our approach represents video clips as nodes in a graph and leverages entities to maintain semantic relationships, thereby enhancing retrieval effectiveness. To address retrieval noise, we proposed a structured query refinement strategy that systematically filters out irrelevant clips, ensuring a more precise selection of relevant video content. Additionally, we introduced an intermediate reasoning step that summarizes the response to the structured query, using the filtered retrieved clips as multimodal context to significantly improve the accuracy of the final answer generation. Our framework outperforms state-of-the-art video RAG methods by 8.6%, demonstrating its effectiveness in enhancing long-video understanding tasks. This work paves the way for more accurate and context-aware long-form video retrieval and reasoning systems.

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

# A  Experiments

## A.1  Category-level performance on MLVU

Table 6 shows performance on the multiple-choice task of MLVU [57]. Our framework consistently improves all models, enhancing LongVU by 5.4% and Qwen2.5VL (7B) by 3.3%. Notably, Vgent achieves 70.4% accuracy on Qwen2.5VL (3B), surpassing its 7B counterpart and improving the base model by 4.2%. Significant gains are observed in Count and Order tasks, highlighting the effectiveness of our approach in cross-segment reasoning and long-video understanding.

Table 6: **Category-level performance on MLVU** [57]. Our framework consistently improves all models, enhancing LongVU by 5.4% and Qwen2.5VL (7B) by 3.3%. Notably, Vgent achieves 70.4% accuracy on Qwen2.5VL (3B), surpassing its 7B counterpart and improving the base model by 4.2%. Significant gains are observed in Count and Order tasks, highlighting the effectiveness of our approach in cross-segment reasoning and long-video understanding.

| Model | Size | Count | Ego | Needle | Order | PlotQA | Anomaly | Topic | Overall |
|---|---|---|---|---|---|---|---|---|---|
| | | | | *Proprietary LVLMs* | | | | | |
| GPT-4o | - | 46.3 | 57.1 | 64.8 | 56.7 | 65.1 | 74.5 | 87.4 | 64.6 |
| | | | | *Open-Source LVLMs* | | | | | |
| InternVL2.5 [8] | 2B | 34.9 | 50.4 | 61.6 | 34.7 | 62.8 | 61.5 | 81.5 | 56.7 |
| InternVL2.5 + Vgent (Ours) | 2B | 59.2 | 53.1 | 66.7 | 38.2 | 63.9 | 62.0 | 81.1 | 61.1$^{+4.4}$ |
| Qwen2-VL [46] | 2B | 30.1 | 56.0 | 72.3 | 32.8 | 65.3 | 55.5 | 80.4 | 58.6 |
| Qwen2-VL + Vgent (Ours) | 2B | 58.7 | 57.6 | 76.9 | 34.3 | 63.8 | 59.5 | 80.3 | 62.5$^{+3.9}$ |
| Qwen2.5-VL [4] | 3B | 36.4 | 53.0 | 77.7 | 55.5 | 70.1 | 75.5 | 86.4 | 66.2 |
| Qwen2.5-VL + Vgent (Ours) | 3B | 60.1 | 58.1 | 78.5 | 61.7 | 70.5 | 74.5 | 87.3 | 70.4$^{+4.2}$ |
| LongVU [54] | 7B | 28.9 | 59.3 | 76.3 | 58.3 | 71.6 | 76.0 | 87.5 | 65.4 |
| LongVU + Vgent (Ours) | 7B | 60.0 | 62.3 | 76.5 | 60.1 | 71.6 | 76.4 | 87.8 | 70.8$^{+5.4}$ |
| Qwen2-VL [46] | 7B | 32.5 | 62.0 | 79.1 | 53.2 | 69.6 | 63.0 | 85.3 | 65.7 |
| Qwen2-VL + Vgent (Ours) | 7B | 60.2 | 65.8 | 80.2 | 60.2 | 70.6 | 63.5 | 86.1 | 70.3$^{+4.6}$ |
| LLaVA-Video [56] | 7B | 42.2 | 61.5 | 76.3 | 61.0 | 75.8 | 72.0 | 85.3 | 69.5 |
| LLaVA-Video + Vgent (Ours) | 7B | 58.7 | 63.0 | 76.9 | 67.1 | 76.4 | 72.5 | 86.9 | 72.5$^{+3.0}$ |
| Qwen2.5-VL [4] | 7B | 41.7 | 58.1 | 78.0 | 61.0 | 73.6 | 72.5 | 87.4 | 68.8 |
| Qwen2.5-VL + Vgent (Ours) | 7B | 58.7 | 59.5 | 79.7 | 67.1 | 74.6 | 74.0 | 88.1 | 72.1$^{+3.3}$ |

## A.2  Confidence-based Refinement

A straightforward solution is to filter out the hard negative retrievals by their relevance scores. Initially, we experimented with confidence-based refinement, as used in VideoAgent [47], where the model self-reflect the relevance of retrieved nodes. However, this approach proved ineffective in our case, as the confidence score failed to reliably reflect video clip relevance, leading to an average improvement of only 0.2%, as shown in Table 7.

Table 7: **Ablation study results** of the performance improvement contributed by each component of our proposed pipeline. CR denotes confidence-based reasoning and SR is our proposed structured reasoning.

| Models | MLVU | VideoMME | LongVideoBench |
|---|---|---|---|
| Qwen2.5-VL [4] | 68.8 | 71.1 | 56.0 |
| Qwen2.5-VL + GraphRAG + CR | 69.5 | 72.9 | 57.5 |
| Qwen2.5-VL + GraphRAG + SR (default) | **72.1** | **74.3** | **59.7** |

### A.3 Baseline Details

**NaïveRAG:** Following GoldFish [3], we construct a NaïveRAG baseline for video understanding by representing each video clip as plain text and retrieving relevant clips based on similarity to the query.

**Video-RAG:** [33]: This method selects keyframes by evaluating CLIP similarity between each frame's features and the text embeddings of keywords extracted from the question. Additionally, an object detection model and an Optical Character Recognition (OCR) model are applied to each keyframe to extract detailed information.

**Proprietary LLM-based:** VideoAgent [47], LLoVi [52], DrVideo [34] and VideoTree [49] utilizes interactive reasoning and planning of proprietary LLM APIs to enhance long-video understanding.

### A.4 Retrieval Embedding

We explore different types of retrieval embeddings, i.e., CLIP [41], BERT [10] and BGE [51] on VideoMME [13] benchmark with Qwen2.5-VL [4] backbone, as shown in Figure 4 (left).

### A.5 Number of Retrieval $N$

We conduct ablation on the number of retrieval $N$ before structured reasoning (SR) on VideoMME [13] benchmark with Qwen2.5-VL [4] backbone, as shown in Figure 4 (middle). We set $N = 20$ by default.

### A.6 Retrieval Threshold $\tau$

We investigate retrieval threshold $\tau$ on VideoMME [13] benchmark with Qwen2.5-VL [4] backbone, as shown in Figure 4 (right). As the value of $\tau$ increases, less video clips are retrieved based on similarity scores, potentially leading to the loss of relevant information. We set $\tau = 0.5$ by default.

### A.7 Qualitative Results

We show a qualitative example in Figure 3, 5 and 6. Our graph construction effectively connects relevant video clips through shared entities. In Figure 3 the graph-based retrieval system can identify relevant nodes that contains a laptop, with Clip 6 providing crucial evidence to answer the query. However, the model incorrectly responded "No" to the question "Did I open the laptop?", presumably due to hard negatives from multiple clips featuring a opened laptop, hallucinating the model to overlook the closed laptop and the action of opening it.

In contrast, with an intermediate reasoning step, we validate each retrieved node with structured subqueries (e.g., "Is there a laptop open?" "Is someone interacting with the laptop?"). This verified information is aggregated to form an enhanced reasoning chain, allowing the model to correctly infer that the laptop was opened, overcoming the distraction from hard negatives.



Figure 4: Ablation studies. **Left**: retrieval embedding. **Middle**: number of retrieval $N$ before SR. **Right**: ablation on retrieval threshold $\tau$.

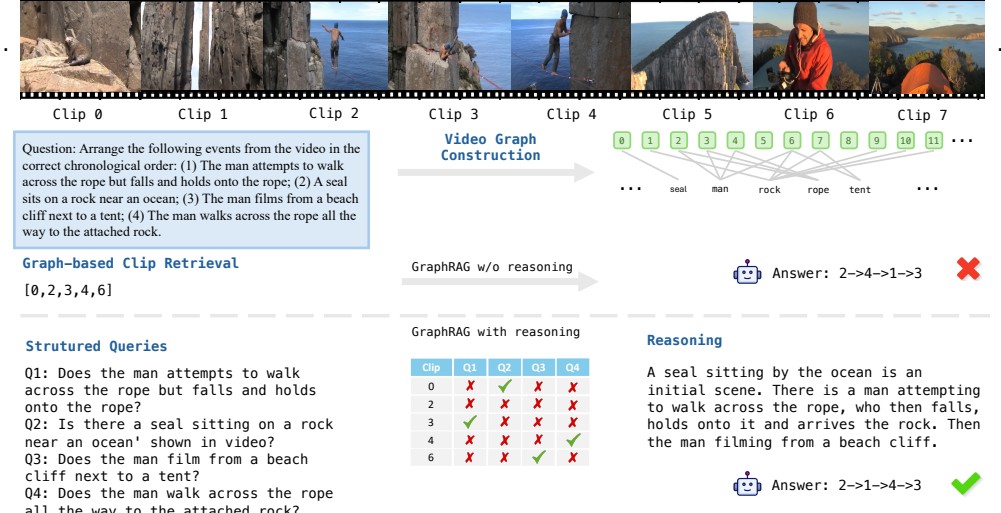

Figure 5: A qualitative example illustrates our graph-based retrieval-reasoning approach.

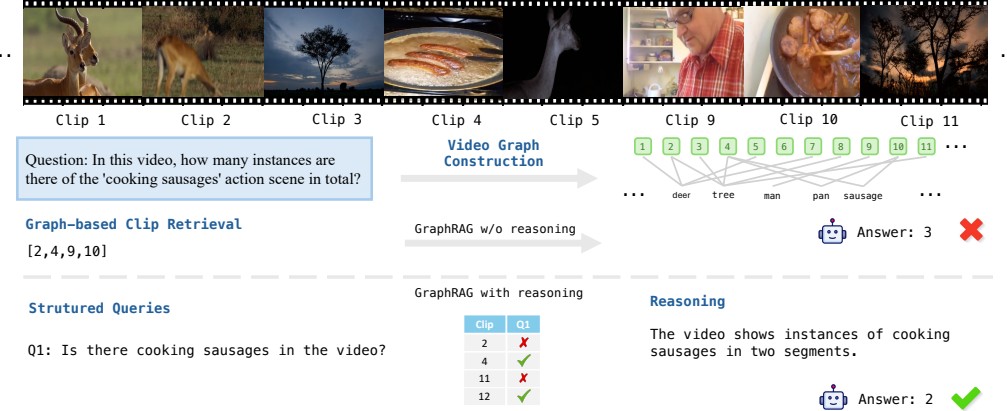

Figure 6: A qualitative example illustrates our graph-based retrieval-reasoning approach.

# B  Prompts

## B.1  Visual Entity Extraction

Figure 7 illustrates the prompts used to describe entities, actions, and scenes given a video segment for the LVLM.

## B.2  Keyword Extraction

Figure 8 presents the prompt designed for the LVLM to perform task identification and extract keywords from the original question to facilitate retrieval.

## B.3  Subqueries Generation

Figure 9 presents the prompt designed for the LVLM to generate structured subqueries for retrieved nodes refinement.

Please analyze the given video and extract key information in a structured JSON format in English. Identify and describe:

**Entities:** List all distinct objects, people, animals, or other significant elements present in the video.

**Actions:** If the entities are interacting, describe their actions and relationships in a structured manner.

**Scenes:** Identify and describe the locations, environments, or contexts where the events occur. If the video is filmed from a first-person point of view, please also describe "subject" as "me" and actions or interactions from this person.

Ensure the output strictly follows the JSON format below:

{ "entities": ["entity name": "", "description": ""], "actions": ["entity name": "action description"], "scenes": ["location": ""] }

The "entity name" in actions should belong to "entity name" in entities.

Each section should be detailed but concise, capturing all relevant interactions and contextual elements from the video. Avoid unnecessary text outside the JSON output.

Figure 7: Prompt for video segment description.

---

**Prompt: Task Identification and Keyword Extraction.**

Given a question of a long video and potential candidates:

**Question:** {query}
**Candidates:** {candidates}

You need to retrieve the relevant video segments to answer the question. Note that you do not need to see the video. But based on the question please think step by step what are the important things for retrieval.

[keywords] Please identify the information, like entities, scene, action from the question that is important to retrieve the segments for further answer the question. Do not include the candidates in the keywords.

[candidates_necessary] Do you think the information in the candidates is necessary for retrieval? Answer yes or no.

[multiple] Do you think it needs to aggregate the information from multiple segments to answer the question? ONLY answer yes or no.

[time] Please identify if it can tell the question is asking which part of the video. Answer begin, end or none.

[tool] Do you think it needs additional step for answering the question, please select from [object counting, action counting, order, none].

[global] Can this question be answered based on the overall understanding of the whole video? (e.g., "What is the main topic of the video?" or "What is the main content of the video?")

Please output the final answer in json format, for example:

{"multiple": "no", "keywords": ["man in black"], "time": "begin", "tool": none, "candidates_necessary": "yes", "global": "yes"}

Figure 8: Prompt for task identification and keywords extraction.

Figure 9: Prompt for subqueries generation.

## C   Model Output Examples

### C.1   Visual Entity Extraction

```
{
    "idx": 0,
    "info": {
        "entities": [
            {
                "entity name": "sailboat",
                "description": "A classic sailboat with white sails
                    and wooden rigging"
            },
            {
                "entity name": "man",
                "description": "A man wearing a dark sweater"
            },
            {
                "entity name": "ocean",
                "description": "A calm ocean under a partly cloudy sky
                    "
            }
        ],
        "actions": [
            {
                "entity name": "sailboat",
                "description": "sailing smoothly on the water"
            },
            {
                "entity name": "man",
                "description": "steering the sailboat"
            },
            {
                "entity name": "man",
                "description": "looking around"
            },
            {
                "entity name": "man",
                "description": "adjusting his hair"
            }
        ],
        "scenes": [
            {
```

```
                "location": "open sea"
            }
        ]
}
```

## D  Limitation

In this work, we represent video content using textual descriptions—such as entities and their associated details—as a lightweight and efficient alternative to raw visual features. However, we do not incorporate visual embeddings or frame-level features into our graph. While computing similarity across frames can be computationally intensive, it remains a promising direction for future improvement.

Additionally, our framework is model-agnostic and compatible with any LVLM, meaning its performance is inherently bounded by the capabilities of the base LVLM. As more powerful LVLMs emerge, our pipeline can be readily adapted to take advantage of their enhanced video understanding and reasoning abilities.

