# OpenReview forum: "Vgent: Graph-based Retrieval-Reasoning-Augmented Generation For Long Video Understanding"
_NeurIPS.cc/2025/Conference — NeurIPS 2025 spotlight_

### Official Review · Reviewer_wVMm · 2025-06-30

**Clarity:** 3
**Significance:** 2
**Originality:** 2
**Rating:** 5
**Confidence:** 3

**Summary:**

This paper presents Vgent, a comprehensive RAG-based framework for long-form video understanding. The key idea is to build the GraphRAG that models video clips as graph nodes, followed by a structured reasoning module, where subqueries are automatically generated to verify and filter noisy content. Extensive experiments on benchmarks like MLVU, LongVideoBench, and VideoMME show that Vgent consistently improves the performance of open-source LVLMs and outperforms prior RAG baselines.

**Questions:**

Some additional questions:

How does Vgent perform without subtitle input?

Is there any special reason not including other datasets such as LVBench and MVBench in the testing?

**Ethical Concerns:**

["NO or VERY MINOR ethics concerns only"]

**Final Justification:**

The authors have well addressed my concerns. The explanation of future work sounds good, especially the incorporation of visual features. The new results in terms of with and without subtitle look reasonable.

**Limitations:**

Yes

**Quality:**

3

**Strengths And Weaknesses:**

Strengths:

The graph-based retrieval module is query-independent and constructed offline, making it efficient and scalable across multi-turn or multi-query QA tasks. Although this feature isn’t particularly novel, it remains essential.

The proposed approach achieves consistent performance improvements over baselines. Ablation studies have also been well done.

The paper is also well structured and clearly presented.

Weaknesses:

The proposed framework improves existing training-free, RAG-based long-form video understanding methods. Although the proposed modules are solid and effective, the overall framework may not have sufficient novelty to be considered a top paper.

The potential and future work of the proposed method are unclear. I may raise my rating if the authors can well address this part.

---

> ### Author Rebuttal · Authors · 2025-07-31
>
> **W1: Although the proposed modules are solid and effective, the overall framework may not have sufficient novelty.**
>
> We respectfully assert the significant novelty of our work, grounded in two key innovations:
> Pioneering cross-clip graph-based retrieval for video understanding: Unlike existing frame-level scene graph methods (e.g., VideoRAG), our approach uniquely connects distinct video segments via shared entities, preserving crucial semantic and contextual relationships across clips.
>
> Then we proposed structured reasoning as a post-retrieval refinement step, a novel, empirically effective mechanism that enhances the accuracy and robustness of retrieved information.
>
> **W2: The potential and future work of the proposed method are unclear.**
> We appreciate the reviewer's query regarding the potential and future work of our proposed method. Our pipeline offers significant advancements across several aspects:
>
> + **Efficient for streaming or long-form videos.** First, we propose an offline graph construction mechanism that provides a structured representation of video segments. This process is performed in a query-independent manner and only once per video. As a result, in multi-round video question answering settings, subsequent queries for the same video can directly retrieve relevant segments without reprocessing the entire video. This significantly improves efficiency, especially for streaming or long-form videos.
> + **Enhanced reasoning in post-retrieval stage.** Second, our proposed structured reasoning further enhances the model reasoning ability. LVLMs often suffer from limited reasoning ability and hallucinations, particularly when faced with hard negative retrievals. Our method addresses this by introducing an intermediate reasoning step to verify retrievals and aggregate information across verified segments, thereby improving the robustness and accuracy of the final response.
>
> In future work, our pipeline can be extended by integrating richer multimodal information into the graph construction process. Due to current computational limitations, we are unable to incorporate visual feature similarity when building the graph. However, with more efficient computation and feature indexing techniques, future versions of the system can leverage visual cues to construct more informative graphs. Additionally, advancements in indexing strategies can further enhance the efficiency and scalability of the retrieval stage, enabling faster and more accurate candidate selection.
>
> **Q1: How does Vgent perform without subtitle input?**
>
> We present below the result for Vgent without subtitle/transcription as input in comparison to base model Qwen2.5-VL and one agent-based baseline Video-RAG.
>
> | Models | Transcriptions / Subtitles | MLVU | VideoMME |
> | ------ | -------------------------- | ---- | -------- |
> | Qwen2.5-VL | No | 68.8 | 65.1 |
> | Qwen2.5-VL + Video-RAG | No | 63.1 | 61.2 |
> | Qwen2.5-VL + Vgent | No | 71.8 | 69.9 |
> | Qwen2.5-VL | Yes | 69.0 | 71.1 |
> | Qwen2.5-VL + Video-RAG | Yes | 63.4 | 65.7 |
> | Qwen2.5-VL + Vgent | Yes | 72.1 | 74.3 |
>
> For VideoMME, questions are designed to necessitate an understanding of both visual and spoken content; consequently, they cannot be answered solely based on visual information. In contrast, MLVU datasets lack dense conversational or spoken content, with their questions primarily focused on visual understanding, independent of audio and spoken information.
>
> **Q2: Is there any special reason not including other datasets such as LVBench and MVBench in the testing?**
>
> MVBench is a short video benchmark, only 16 seconds on average. Our method is specifically designed for long video understanding, operating by segmenting videos into 32-second chunks for graph construction, retrieval, and reasoning. Applying our pipeline to MVBench would effectively reduce it to a naive baseline, as such short videos do not require retrieval or long-range reasoning. Our pipeline is more beneficial in scenarios that require understanding and reasoning over extended video content. As for LongVideoBench, we have presented our model's performance in comparison to the base model Qwen2.5-VL in Table 3 of the main paper.

---

> > ### Comment · Reviewer_wVMm · 2025-08-05
> >
> > Thanks for the rebuttal. I will update my rating accordingly.

---

### Official Review · Reviewer_6nEZ · 2025-07-01

**Clarity:** 4
**Significance:** 3
**Originality:** 4
**Rating:** 5
**Confidence:** 5

**Summary:**

The paper proposes Vgent, a novel graph-based RAG framework designed to improve long video understanding in Large Video Language Models. By representing videos as structured graphs and incorporating an intermediate reasoning step with information aggregation, Vgent addresses key challenges in long-context retrieval and reasoning. Experimental results on three main long-video benchmarks demonstrate consistent performance gains over base models and improvement over prior RAG-based methods.

**Questions:**

1.	In both the entity merging and clip retrieval stages, the similarity is computed based solely on the text embeddings of entity names. How does the method address issues of coreference and ambiguity (e.g., when two different people are both referred to as “a man”)? A more detailed discussion on this would be helpful.

2.	Building an accurate graph from video content is inherently challenging. Has there been any deeper analysis of the quality of the constructed graphs, including ablation studies on parameters like τ?

3.	Since graph construction is likely to be computationally intensive, a time analysis of this component would be valuable for understanding the overall efficiency.

4.	Some implementation details could be clarified. In Table 1, for the different LVLM+Vgent settings, was the graph constructed using the corresponding LVLM in each case? Also, what were the inference-time model settings used in these experiments? It would be good to report them.

**Ethical Concerns:**

["NO or VERY MINOR ethics concerns only"]

**Final Justification:**

All my conerns have been addressed. So  I am inclined to accept the paper.

**Limitations:**

yes

**Quality:**

3

**Strengths And Weaknesses:**

Strengths:

⦁	The proposed method of representing video segments as graph nodes is inspiring, enabling the construction of structured connection between video clips.

⦁	The divide-and-conquer strategy employed in the Retrieval-Reasoning Augmented Generation effectively refines the retrieved video clips, which alleviates the limitations of LVLMs in graph construction and reasoning. This approach is intuitive and makes sense.

⦁	The paper conducts extensive experiments on three widely used long-video benchmarks, with very detailed discussion on the results. The proposed method consistently improves performance across major LVLMs and outperforms existing approaches, demonstrating both effectiveness and generalizability.

Weaknesses:

⦁	The paper lacks a more in-depth analysis of the Offline Graph, including its construction efficiency, accuracy, and both qualitative and quantitative evaluations.

---

> ### Author Rebuttal · Authors · 2025-07-31
>
> **W1 & Q2: The paper lacks a more in-depth analysis of the Offline Graph, including its construction efficiency, accuracy, and both qualitative and quantitative evaluations.**
>
> We have conducted a comprehensive analysis of the Offline Graph's impact and performance through extensive ablations and qualitative examples in our supplementary material. Specifically, we provide detailed quantitative evaluations in the supplementary material covering:
>
> + The impact of different retrieval embedding choices (A.5).
> + The optimal number of retrieved items ($N$) (A.6).
> + The effectiveness of the retrieval threshold ($\tau$) (A.7).
> + Furthermore, our analysis includes significant qualitative evaluations, with several illustrative examples presented in A.8 and Figures 1, 6, and 7.
>
> **Q1: In both the entity merging and clip retrieval stages, the similarity is computed based solely on the text embeddings of entity names. How does the method address issues of coreference and ambiguity (e.g., when two different people are both referred to as “a man”)? A more detailed discussion on this would be helpful.**
>
> We appreciate the reviewer's insightful question regarding coreference and ambiguity in entity merging and clip retrieval. It is important to clarify that our method does not solely rely on entity names for similarity computation.
>
> As shown in Equation 2, we maintain an entity-description pair $(e_j^i​,t_j^i​)$ for each entity $j$ within video clip $V_i$​. The similarity score for merging is then computed between the textual description $t_j^i$​ and the descriptions of existing entities in our global entity set $U$, based on their respective text embeddings. We prompt the model to generate detailed descriptions for each entity, thereby mitigating coreference and ambiguity issues by enriching the textual context with more distinctive and informative cues.
>
> We also explored incorporating visual features of entities into the graph construction process. Specifically, we employed the Florence-2-large model to generate detailed descriptions and bounding boxes for each detected entity. Using these, we cropped the corresponding regions and extracted visual features for integration into the graph. However, this approach introduced substantial computational overhead during both feature extraction and node merging. Moreover, our preliminary experiments showed only marginal performance improvements, i.e., 0.6% on MLVU. As a result, we did not include this design in the final version of our pipeline. We consider multimodal graph construction a promising direction for future work.
>
> **Q3: Time analysis of graph construction.**
>
> We show the inference time analysis of graph construction in Table 4 in the main paper. The offline, query-independent graph construction requires 20.13 seconds to process every minute of video on average.
>
> | Model / Time(sec) | Graph Construction (offline) | Retrieval, Reasoning (online) |
> | ----------------- | ---------------------------- | ----------------------------- |
> | VideoAgent        | N/A                          | 67.25                         |
> | Qwen2.5VL-7B      | N/A                          | 2.95                          |
> | Video-RAG         | N/A                          | 20.81                         |
> | Vgent (Ours)      | 20.13                        | 3.93                          |
>
> **Q4: Was the graph constructed using the corresponding LVLM in each case? What were the inference-time model settings used in these experiments?**
>
> Yes, we use the same model for graph construction and question answering. Regarding the inference-time model settings, we adhered to the configurations (context size, number of frames) specified by each corresponding base model, and reproduced the matched results as reported in their original papers.

---

> > ### Comment · Reviewer_6nEZ · 2025-08-06
> >
> > Thank you for your reply. I am inclined to accept the paper.

---

### Official Review · Reviewer_FuXV · 2025-07-03

**Clarity:** 3
**Significance:** 3
**Originality:** 3
**Rating:** 5
**Confidence:** 4

**Summary:**

This paper proposes Vgent, a novel graph-based retrieval-reasoning-augmented generation framework that enhances open-source Long Video Language Models (LVLMs) in long video understanding tasks. Vgent constructs a Video Knowledge Graph with semantic relationships across video clips preserved to improve retrieval effectiveness and introduces Structured Query Refinement (SQR) to perform intermediate reasoning and information aggregation. Evaluations on three popular benchmarks (MLVU, VideoMME, LongVideoBench) according to Table 1 demonstrate consistent performance improvements across various LVLM backbones.

**Questions:**

- Were the same subtitle/transcript settings applied to all baseline models in Table 1 and 2? How would Vgent perform under *pure visual settings* (e.g., VideoMME-w/o subs or visual-only MLVU)?
- Could the proposed cross-clip graph reasoning approach be further validated on benchmarks focusing specifically on **cross-segment temporal reasoning**?
- What is the upper limit of video frame length that Vgent can handle with its graph-based retrieval structure? Can it scale to hour-long videos? What are the average processing times on VideoMME-long and LVBench?
- Appendix Figure 5 suggests that Top-N selection has limited impact on VideoMME performance—does the same trend hold on other datasets like MLVU and LongVideoBench?

**Ethical Concerns:**

["NO or VERY MINOR ethics concerns only"]

**Final Justification:**

The authors’ rebuttal satisfactorily addresses my concerns, particularly the comparison of results with and without subtitles. I recommend incorporating these findings into the revision, and I am inclined to raise my score.

**Limitations:**

The authors discuss limitations in detail in Appendix D.

**Quality:**

3

**Strengths And Weaknesses:**

### **Strengths:**

1. The proposed training-free graph-based retrieval and reasoning framework eliminates dependency on proprietary models or large external backbones, providing a flexible and self-contained solution.
2. Vgent achieves consistent improvements on three widely-used long video understanding benchmarks (MLVU, VideoMME, LongVideoBench), and an experimental comparison is made with recent work on similar graph structure methods such as Video-RAG.

### **Weaknesses:**

- **Concerns about fair comparison.** As stated in lines 236–238, the authors used `openai/whisper-large` for spoken content extraction in MLVU and used benchmark-provided subtitles for VideoMME and LongVideoBench. It is unclear whether the same preprocessing pipeline was applied in comparison baselines  in Table 1 and 2(especially Table 2, e.g., “Qwen2.5-VL + Video-RAG” and “LLaVA-Video + Video-RAG”). Moreover, Table 1 reports results for LongVU on MLVU that align with the original paper, while the LLaVA-Video results differ. Many existing long video models (e.g., LongVA, LongVILA, VideoChat-flash, Apollo) do not include audio-derived transcripts during evaluation, as such textual tokens may heavily influence model performance. A comparison under *pure visual settings* with SOTA methods would better validate the visual reasoning capability of Vgent.
- The proposed **Structured Query Refinement (SQR)** involves decomposing a query into multiple sub-questions and validating against multiple retrieved clips with VLM. While this increases accuracy, it may introduce significant **inference latency**, especially when scaling to longer videos or larger candidate sets.
- The paper lacks ablation studies on critical hyperparameters such as **Top-N retrieval size**. The choice of N directly impacts whether the reasoning module can access necessary clips for answering decomposed binary questions. More discussions are needed on how N influences model performance and efficiency trade-offs.

---

> ### Author Rebuttal · Authors · 2025-07-31
>
> **W1 (1) & Q1: Concerns about fair comparison for using video subtitles.**
>
> For evaluations on both VideoMME and LongVideoBench, all models utilize the subtitles provided by the respective benchmarks, ensuring a fair and consistent comparison across all approaches. For MLVU, we extracted subtitles to ensure consistent, fair comparisons with other agent-based or tool-using methods like Video-RAG.
>
> **W1 (2) : Moreover, Table 1 reports results for LongVU on MLVU that align with the original paper, while the LLaVA-Video results differ.**
>
> Our reproduced result for LLaVA-Video matched their reported result in the original paper.
>
> | Models | MLVU | VideoMME |
> | -------- | ------- | ------- |
> | LLaVA-Video-7B (reported in original paper)  | 70.8 | 69.7 |
> | LLaVA-Video-7B (reproduced) | 69.5 | 69.2 |
>
> **W1 (3): Many existing long video models (e.g., LongVA, LongVILA, VideoChat-flash, Apollo) do not include audio-derived transcripts during evaluation, as such textual tokens may heavily influence model performance.**
>
> For VideoMME, most of the models mentioned above (LongVILA, VideoChat-Flash, Apollo) reported the performance with subtitles. Since questions in VideoMME require the model to understand both visual and spoken content, those questions could not be answered solely on visual information. Unlike VideoMME, MLVU does not contain dense conversational or spoken content. The questions in MLVU are primarily focused on visual understanding and are independent from audio and spoken information.
>
> **W1 (4): A comparison under pure visual settings with SOTA methods would better validate the visual reasoning capability of Vgent.**
>
> We appreciate the reviewer's suggestion for a comparison under pure visual settings to validate visual reasoning capabilities. We provided the result comparing performance with and without transcriptions/subtitles below.
>
> | Models | Transcriptions / Subtitles | MLVU | VideoMME |
> | ------ | -------------------------- | ---- | -------- |
> | Qwen2.5-VL | No | 68.8 | 65.1 |
> | Qwen2.5-VL + Video-RAG | No | 63.1 | 61.2 |
> | Qwen2.5-VL + Vgent | No | 71.8 | 69.9 |
> | Qwen2.5-VL | Yes | 69.0 | 71.1 |
> | Qwen2.5-VL + Video-RAG | Yes | 63.4 | 65.7 |
> | Qwen2.5-VL + Vgent | Yes | 72.1 | 74.3 |
>
> **W2: While SR increases accuracy, it may introduce significant inference latency, especially when scaling to longer videos or larger candidate sets.**
>
> We respectfully disagree with this statement. Structured Reasoning (SR) is applied only after the Top-$N$ retrieval, meaning it processes at most $N$ segments. As a result, its computational cost does not scale with the length of the video and does not introduce additional latency for longer inputs.
>
> **W3: The paper lacks ablation studies on critical hyperparameters such as Top-N retrieval size. The choice of N directly impacts whether the reasoning module can access necessary clips for answering decomposed binary questions. More discussions are needed on how N influences model performance and efficiency trade-offs.**
>
> We appreciate the reviewer's comment regarding ablation studies on critical hyperparameters. We have indeed conducted a detailed ablation study on the Top-$N$ retrieval size, as presented in A.6 of our supplementary material (zip file). The results, also summarized below, illustrate the impact of $N$:
>
> | N    | Short | Medium | Long |
> | ---- | ----- | ------ | ---- |
> | 5    | 76.5  | 74.2   | 61   |
> | 10   | 76.8  | 75.3   | 64.1 |
> | 15   | 77.1  | 77.3   | 66.5 |
> | 20 (default)   | 77.2  | 78.0  | 67.7 |
> | 25   | 77.2  | 78.1   | 67.9 |
>
> As shown, increasing the retrieval size from $N=5$ to $N=20$ consistently leads to more optimal performance, particularly noticeable on long videos. This improvement stems from providing greater tolerance to embedding search errors, ensuring the reasoning module can access necessary clips. Furthermore, potential errors in the Top-$N$ retrieved segments are mitigated through structured reasoning, which effectively filters out retrieval noise. Beyond $N=20$, further increases in size introduce only marginal gains while raising computational costs, indicating that most relevant segments are already retrieved within this range.
>
> **Q2: Could the proposed cross-clip graph reasoning approach be further validated on benchmarks focusing specifically on cross-segment temporal reasoning?**
>
> Table 2 in the supplementary material presents category-level performance on MLVU. The Order category represents a cross-segment temporal reasoning task, involving questions that require understanding the order of different actions across multiple video segments. *(We post one example here: "Arrange the following events from the video in the correct chronological order: (1) Woman tapes her hands with white tape; (2) Woman starts boxing in the ring with a guy; (3) Woman does sit ups on a towel on the beach; (4) Pictures of woman in her bikini are shown.")*. In this challenging category, our pipeline achieves a 6.1% improvement over the base model Qwen2.5-VL, demonstrating its effectiveness in temporal reasoning.
>
> **Q3 (1): What is the upper limit of video frame length that Vgent can handle with its graph-based retrieval structure? Can it scale to hour-long videos?**
>
> Vgent's design, utilizing a graph-based retrieval structure, offers unlimited scalability regarding video frame length, making it fully capable of handling hour-long videos and beyond. The core strength lies in its ranking-based retrieval mechanism, which efficiently identifies and prioritizes the most relevant video segments irrespective of the overall video's duration. This capability is empirically validated by our results on the VideoMME benchmark's long split, where video durations range from 30 to 60 minutes. As presented in Table 1 of our main paper, Vgent significantly outperforms the base model in answer accuracy for long video scenarios.
>
> **Q3 (2): What are the average processing times on VideoMME-long and LVBench?**
>
> For VideoMME's long split (videos ranging from 30 to 60 minutes), the average processing time is 297.4 seconds. It's important to note that our graph construction is query-independent. Since each VideoMME video typically has three associated questions, the graph is built only once per video, significantly amortizing this cost across multiple queries. For LongVideoBench, the average processing time is 182.8 seconds.
>
> **Q4: Appendix Figure 5 suggests that Top-N selection has limited impact on VideoMME performance—does the same trend hold on other datasets like MLVU and LongVideoBench?**
>
> We appreciate the reviewer’s observation. The perceived limited impact in Figure 5 may be due to scaling issues in the visualization. As shown in the table above, Top-N retrieval actually has a significant influence on VideoMME performance. Specifically, increasing the number of retrieved candidates from $N=5$ to $N=20$ improves the accuracy from 61.1% to 67.7% on VideoMME long split, representing a notable gain of 6.6%. We observe a similar trend on other datasets such as MLVU and LongVideoBench, where larger $N$ values generally lead to improved retrieval quality and downstream performance.
>
> | N  | MLVU | LongVideoBench | VideoMME (long) |
> | ---- | ----- | ------ | ---- |
> | 5    | 70.2  | 57.3   | 61 |
> | 20   | 72.1 | 59.7 | 67.7 |

---

> > ### Comment · Reviewer_FuXV · 2025-08-06
> >
> > Thank the authors for their detailed rebuttal, which satisfactorily addresses my concerns, particularly the comparison of results with and without subtitles. I recommend incorporating these findings into the revision, and I am inclined to raise my score.

---

### Official Review · Reviewer_3ccS · 2025-07-03

**Clarity:** 4
**Significance:** 3
**Originality:** 4
**Rating:** 4
**Confidence:** 4

**Summary:**

It introduces the Vgent, a graph-based Retrieval-Augmented Generation (RAG) framework tailored for long video understanding. The method addresses the inherent challenges of applying RAG in the video domain, such as disrupted temporal dependencies and noise from irrelevant content. This paper proposes a novel pipeline that includes (i) offline video graph construction to preserve semantic and temporal relationships across clips, and (ii) a structured post-retrieval reasoning step that filters retrieved clips and aggregates relevant information for more accurate responses. The Vagent framework demonstrates consistent improvements (3.0%–5.4%) over base open-source LVLMs and outperforms state-of-the-art RAG-based methods by 8.6% across three long-video benchmarks.

**Questions:**

1. This work selects the clips according to the embedding similarity, which might have problems of false-positive and true-negative. The method introduces structured query refinement to try to filter the false-positive question. So how the framework resists the true-negative problem if some key information is lost in the early stage of the pipeline?

2. Have you considered non-entity-based or multi-modal graph construction (e.g., action sequences, visual similarity)? Would incorporating motion features (e.g., optical flow) improve temporal coherence?

3. In Section 4.3, the Ablation Studies section, you show that adding Structured Reasoning (SR) shows more improvement than the GraphRAG module, showing the importance of the SR approach, but the source of this improvement remains somewhat abstract. Have you conducted a finer-grained analysis on where SR contributes the most? This is what i am very curious.

**Ethical Concerns:**

["NO or VERY MINOR ethics concerns only"]

**Final Justification:**

I am overall satisfied with the author rebuttal.

**Limitations:**

yes

**Quality:**

3

**Strengths And Weaknesses:**

Strengths:

1. This paper provides a quite solid and interesting method. It combines graph-based representation and middle-stage reasoning into one pipeline, which doesn’t need extra training. The idea is fresh and practical.

2. The method is introduced clearly, and the diagrams are easy to understand. The whole paper has good structure and is not difficult to read.

3. The problem of long video understanding is very meaningful. Especially now, large vision-language models have some trouble when facing long input. The results in experiments show that Vgent improves both retrieval accuracy and reasoning ability a lot.

4. Although this work is based on some previous ideas like RAG and scene graph, the usage of offline entity video graph and structured sub-query checking is new and not so simple. It shows the authors did deep thinking.

Weaknesses:

1. It uses multiple calls for LVM tools, and there is accumulated errors in the graph construction and filtering process.

2. The novelty is not limited compared to the existing scene-graph based VLLM works. the innovation lies it introduces a problem decomposition and verification step it is good but not so novel.

3. Limited ablation on graph construction strategy: The authors assume the effectiveness of entity-level similarity and merging, but do not explore alternative graph construction techniques, like, motion cues, which is very important in some queries where dynamic information in videos are needed.

4. The method relies heavily on accuracy improvements, but lacks deeper analysis on error types, qualitative improvements, or user studies for interpretability gains.

---

> ### Author Rebuttal · Authors · 2025-07-31
>
> **W1 (1): It uses multiple calls for LVM tools**
>
> The utilization of multiple LVM calls is inherent to the design of sophisticated agent-based approaches, a common practice, not a weakness, in the field. For example, VideoTree [1] called LLM for relevance scoring by breadth and depth expansion. DrVideo [2] iteratively searches videos with an agent-based loop to converge on final predictions. Multiple calling should not be our weakness. Furthermore, our refinement and answer stages operate on a significantly reduced set of videos, N=20 after retrieval and r=5 after filtering, which significantly reduces computations.
>
> [1] VideoTree: Adaptive Tree-based Video Representation for LLM Reasoning on Long Videos, CVPR 2025
>
> [2] DrVideo: Document Retrieval Based Long Video Understanding, CVPR 2025
>
> **W1 (2): There is accumulated errors in the graph construction and filtering process.**
>
> We respectfully disagree with the claim that errors accumulate during graph construction and filtering.
> Naïve RAG can indeed suffer from information loss by summarizing video segments into textual description, however, we try to preserve information in a structured manner through graph construction. By saving entities along with corresponding attributes and actions, we alleviate the information loss and error accumulation issue compared to naïve text-based retrieval. To validate this, we evaluated our method on the temporal grounding benchmark ReXTime [1], measuring both retrieval performance (Recall@1 with IoU > 0.3) and final answer accuracy. As shown in the table below, our method (Vgent) outperforms the NaïveRAG baseline by a significant margin in both metrics:
>
> | ReXTime | R@1 (IoU>0.3) | Acc |
> | ----- | ----- | ----- |
> | NaïveRAG | 23.6 | 30.1 |
> | Vgent | 27.8 | 35.6 |
>
> Moreover, our filtering process reduces retrieval noise by eliminating irrelevant information during the retrieval stage, retaining only the most relevant video segments necessary for the final answer. This effectively alleviates potential error accumulation introduced during retrieval. Consequently, the integration of Structured Reasoning (SR) yields an average improvement of 2.2% across three benchmarks.
>
> | Methods | MLVU | VideoMME | LongVideoBench |
> | ----- | ----- | ----- | ----- |
> | Qwen2.5-VL + GraphRAG | 69.5 | 72.7 | 57.1 |
> | Qwen2.5-VL + GraphRAG + SR | 72.1 | 74.3 | 59.7 |
>
> [1] ReXTime: A Benchmark Suite for Reasoning-Across-Time in Videos, NeurIPS 2024
>
> **W2: The novelty is not limited compared to the existing scene-graph based VLLM works. the innovation lies it introduces a problem decomposition and verification step it is good but not so novel.**
>
> We respectfully disagree with the comment on limited novelty. Our work introduces two distinct and significant innovations:
> + First, we've introduced the idea of cross-clip graph-based retrieval for video understanding. This is a novel approach that differentiates us from existing scene-graph based VLLM methods. For instance, approaches like VideoRAG are limited to frame-level scene graphs, neglecting the crucial relationships that span across frames or clips. However, we innovate by connecting distinct video segments via shared entities, which preserves semantic relationships and contextual dependencies.
> + Second, it is not just simple verification. One naive way is just to let the model verify whether each retrieved video segments are relevant or not. We designed the structured reasoning as a post-retrieval refinement step. This mechanism, designed to enhance the accuracy and robustness of retrieved information, is both novel and empirically effective, which has not been explored in existing literature.
>
> **W3 & Q2 (1): Limited ablation on graph construction strategy: The authors assume the effectiveness of entity-level similarity and merging, but do not explore alternative graph construction techniques, like, motion cues, which is very important in some queries where dynamic information in videos are needed.**
>
> We appreciate the reviewer's excellent point regarding the exploration of alternative graph construction techniques and the incorporation of multimodal information. Our approach is not exclusively limited to static entity names. As detailed in Appendix C.1, our visual entity extraction process also includes actions, providing a level of dynamic information.
>
> **W3 & Q2 (2): Have you considered non-entity-based or multi-modal graph construction (e.g., action sequences, visual similarity)?**
>
> We have explored incorporating visual features of entities into the graph construction process. Specifically, we employed the Florence-2-large model to generate detailed descriptions and bounding boxes for each detected entity. Using these, we cropped the corresponding regions and extracted visual features for integration into the graph. However, this approach introduced substantial computational overhead during both feature extraction and node merging. Moreover, our preliminary experiments showed only marginal performance improvements, i.e., 0.6% on MLVU. As a result, we did not include this design in the final version of our pipeline. We consider multimodal graph construction a promising direction for future work.
>
> **W3 & Q2 (3): Would incorporating motion features (e.g., optical flow) improve temporal coherence?**
>
> Regarding more dense and low-level motion cues like optical flow, we recognize their potential. However, computing optical flow is very computationally intensive, especially for long videos, which is not a primary focus of our work. We agree that temporal motions derived from optical flow could be beneficial in specific, action-focused benchmarks that demand fine-grained dynamic understanding. This remains a promising avenue for future work.
>
> **W4: The method relies heavily on accuracy improvements, but lacks deeper analysis on error types, qualitative improvements, or user studies for interpretability gains.**
>
> We've conducted significant qualitative analysis of the enhancements derived from both our graph construction and structured reasoning components. Specifically, we've investigated the qualitative improvements resulting from both our graph construction and structured reasoning components. Detailed examples and visual evidence illustrating these gains can be found in A.8 and Figures 1, 6, and 7 of our supplementary material (PDF in the zip file). Furthermore, Table 6 presents category-level performance on MLVU, investigating how the proposed method gains across specific tasks. Notably, we observe substantial gains in action counting and ordering tasks, highlighting the effectiveness of our approach in cross-segment temporal reasoning.
>
> | Model              | Count | Order |
> | ------------------ | ----- | ----- |
> | Qwen2.5-VL         | 41.7  | 61.0  |
> | Qwen2.5-VL + Vgent | 58.7  | 67.1  |
>
> **Q1: This work selects the clips according to the embedding similarity, which might have problems of false-positive and true-negative. The method introduces structured query refinement to try to filter the false-positive question. So how the framework resists the true-negative problem if some key information is lost in the early stage of the pipeline?**
>
> We assume that the reviewer’s concern about the “true-negative” problem is a question of whether key information might be lost during the retrieval stage prior to structured reasoning.
>
> Our ranking-based retrieval approach is designed to tolerate retrieval errors (true-negative problem). And by adjusting retrieval threshold ($\tau$) and the number of retrieved candidates (N), we effectively minimize information loss. We conduct ablation studies on the $\tau$ and $N$, as shown in Figure 5 (in the supplementary PDF within the zip file). The results show that increasing the retrieval size ($N = 20$) leads to optimal performance, particularly on the long-video split. This improvement stems from a larger candidate set to retain more relevant information. Beyond $N=20$, further increases in $N$ yield only minimal additional enhancement.
>
> | N    | Short | Medium | Long |
> | ---- | ----- | ------ | ---- |
> | 5    | 76.5  | 74.2   | 61   |
> | 10   | 76.8  | 75.3   | 64.1 |
> | 15   | 77.1  | 77.3   | 66.5 |
> | 20   | 77.2  | 78.0  | 67.7 |
> | 25   | 77.2  | 78.1   | 67.9 |
>
> Subsequently, our structured refinement module effectively addresses the false-positive problem by filtering out irrelevant or noisy retrievals.
>
> **Q3: Why adding Structured Reasoning (SR) shows more improvement than the GraphRAG module? Finer-grained analysis on where SR contributes the most?**
>
> Since direct ground truth for retrieval accuracy is unavailable, we manually examined both the retrieved nodes and the model's responses for some examples. We observed, in many cases, the correct video segment needed for a question was indeed retrieved, yet the model produced an incorrect answer. However, when the model was given only this accurate segment, it often yielded the correct response, indicating that retrieval noise (ranking-based Top N retrieval) significantly contributes to prediction errors.
>
> This explains why relying solely on graph-based retrieval offered only marginal performance gains, and why structured reasoning played a more critical role.
>
> In A.8 of our supplementary materials (pdf in zip file), we present several fine-grained cases that demonstrate how GraphRAG, when combined with structured queries and reasoning, leads to accurate answers. For instance, Figure 6 in the supplementary materials shows that, w/o SR, the model failed to correctly answer an action ordering question based on the retrieved segments. However, by verifying and filtering out irrelevant segments and incorporating an intermediate reasoning step to summarize the sequence of events, the model successfully produces the correct answer.

---

> > ### Author Response · Authors · 2025-08-07
> >
> > Dear Reviewer 3ccS,
> >
> > As the discussion period is nearing its end, we'd like to make sure we've satisfactorily addressed all your points. We truly value your insights, so if there's anything else you'd like to bring up or any remaining feedback you have, please let us know.
> >
> > Thank you for your time and effort in reviewing our paper.

---

> > ### Comment · Reviewer_3ccS · 2025-08-07
> >
> > Thank you for the response, I will raise my score.

---

### Comment · Area_Chair_oQqK · 2025-08-06
**Reminder of Submitting Final Score**

Dear Reviewers,

This is a gentle reminder that the authors added their response on your question. Can you provide your feedback on their response? Please keep in mind that the deadline of August 6th approaching, and your additional timely feedback greatly enhance further discussions if needed.

Best regards
AC

---

### Decision · Program_Chairs · 2025-09-17

**Decision:**

Accept (spotlight)

**Comment:**

The paper presents Vgent, a graph-based retrieval-reasoning framework for long video understanding, which builds offline video graphs to preserve semantic/temporal relations and introduces structured query refinement for noise reduction and better reasoning. It is training-free, flexible, and reproducible, and demonstrates consistent and notable gains (3–5% over LVLM baselines, ~8% over prior RAG methods) across three standard benchmarks. The strengths lie in tackling an important and timely challenge, providing a practical and well-structured solution, and showing robust improvements across multiple LVLMs. While some concerns remain about incremental novelty, reliance on subtitles, and limited graph analysis, the rebuttal clarified comparisons and scalability, convincing reviewers that the method is technically solid and impactful. Given its clear contributions to a high-priority problem area, strong empirical evidence, and broad applicability, I recommend acceptance as a spotlight, recognizing its significance in advancing long video reasoning in LVLMs.